

# miR-29a-3p directly targets Smad nuclear interacting protein 1 and inhibits the migration and proliferation of cervical cancer HeLa cells

Ying Chen[1], Weiji Zhang[1], Lijun Yan[1], Peng Zheng[2] and Jin Li[1]

[1] College of Life Science, Yangtze University, Jingzhou, Hubei, China
[2] Institute of Biology and Medicine, College of Life Science and Healthy, Wuhan University of Science and Technology, Wuhan, Hubei, China

## ABSTRACT

Smad nuclear interacting protein 1 (SNIP1) is a nuclear protein and involved in essential biological processes. MicroRNAs are effective regulators of tumorigenesis and cancer progression via targeting multiple genes. In present study, we aimed to investigate the function of SNIP1 and identify novel miRNA-SNIP1 axis in the development of cervical cancer. The results showed for the first time that silencing of the *SNIP1* gene inhibited the migration and proliferation in HeLa cells significantly. Bioinformatics analysis and dual luciferase reporter assay demonstrated that miR-29a-3p could target 3′ UTR of SNIP1 directly. The mRNA and protein expression levels of SNIP1 were negative regulated by miR-29a-3p according to the RT-qPCR and Western blot analysis, respectively. Furthermore, functional studies showed that over-expression of miR-29a-3p restrained HeLa cells migration and proliferation, and the mRNA expression of SNIP1 downstream genes (*HSP27*, *c-Myc*, and *cyclin D1*) were down-regulated by miR-29a-3p. Together, we concluded that miR-29a-3p suppressed the migration and proliferation in HeLa cells by directly targeting SNIP1. The newly identified miR-29a-3p/SNIP1 axis could provide new insight into the development of cervical cancer.

# INTRODUCTION

Cervical cancer is the fourth most prevalent malignancy and remains the major cause of cancer death among females worldwide (*Bray et al., 2018*). Despite at there are considerable improvements of treatment strategies in the therapy of cervical cancer, the overall 5-year survival rates of patients remain poor for metastasis (*Forouzanfar et al., 2011*; *Tewari et al., 2014*; *Li, Wu & Cheng, 2016*). Therefore, understanding the pathogenesis and progression of cervical cancer is quite necessary and may facilitate identification of effective therapeutic targets for cervical cancer treatment.

Smad nuclear interacting protein 1 (SNIP1), as an evolutionarily conserved nuclear protein, is involved in essential biological processes such as cell proliferation (*Roche et al., 2004*; *Fujii et al., 2006*), small RNA biogenesis (*Yu et al., 2008*), DNA damage response (*Chen et al., 2018*), and several signaling pathways (*Kim et al., 2000*; *Kim et al., 2001*). In

Corresponding authors
Ying Chen,
chenyingcy@yangtzeu.edu.cn
Jin Li, lijin28@yangtzeu.edu.cn

patients with non-small cell lung cancer or tongue squamous cell carcinoma, SNIP1 might be a reliable prognostic indicator (*Liang et al., 2011*; *Jeon et al., 2013*). Recently, SNIP1 was considered to be targeted by microRNA-335 and involved in osteosarcoma proliferation and metastasis (*Xie et al., 2019*). However, the exact role of SNIP1 in the development of cervical cancer remains obscure.

MicroRNAs (miRNAs) are a group of non-coding RNAs with 19-25 nucleotides and have closely relationships with the occurrence and progression of human cancers (*Romano et al., 2017*). MicroRNAs regulate gene expression mainly through binding to the 3′-untranslated region (UTR) of target mRNAs (*Bartel, 2004*). As oncogene or tumor suppressor in different cancers, miRNAs play essential roles in basic biological processes such as cell proliferation, apoptosis, differentiation, migration and invasion (*Bartel, 2004*; *Pardini et al., 2018*). Accumulating studies have shown that miR-29a was abnormally expressed in various cancers, including cervical cancer (*Pei, Lei & Liu, 2016*; *Yang et al., 2017*; *Gong et al., 2019*). However, the detailed role and underlying mechanism of miR-29a in cervical cancer is still largely unclear.

In this research, we investigated the biological role of SNIP1 in the progression of cervical cancer. Furthermore, we predicted and demonstrated that miR-29a-3p inhibited the transcription and protein expression of SNIP1 by targeting 3′ UTR directly, and suppressed the proliferation and migration of cervical cancer cells.

## MATERIALS & METHODS

### Cell culture
The human cervical cancer HeLa cells were purchased from American Type Culture Collection (Manassas, VA), and maintained in Dulbecco's modified Eagle's medium (Hyclone, USA) containing 10% fetal bovine serum (Gibco, USA) under a humidified environment with 5% $CO_2$ at 37 °C.

### Transfection
Three small interfering RNAs (siRNAs) targeting SNIP1 (siSNIP1-330, siSNIP1-871, siSNIP1-1059) were purchased from GenePharma (Suzhou, China). The miR-29a-3p mimics and negative control (NC) were manufactured by RiboBio (Guangzhou, China). All transfections were performed using siRNA-Mate reagent (GenePharma, China) in accordance with the instruction manual. The cells were collected after 48 h of transfection for further experiments.

### Quantitative RT-PCR (RT-qPCR)
Total RNA was harvested from HeLa cells by EZNA Total RNA Kit (Omega BioTek, USA). The first-strand cDNA was generated using HiScript II Q RT SuperMix (Vazyme, China). RT-qPCR was performed to quantify relative RNA levels using ChamQ Universal SYBR qPCR Master Mix (Vazyme, China) on a CFX96 Touch (Bio-rad, USA). The $2^{-\Delta\Delta Ct}$ method was used to measure the relative expression level, and GAPDH served as the internal reference. The primers used for RT-qPCR were presented in Table 1.

**Table 1  Primers for quantitative RT-PCR.**

| Gene | Forward 5′–3′ | Reverse 5′–3′ |
|---|---|---|
| SNIP1 | GCTTTGTGGACCAGGTGTTT | TGTACAGTCACGGGCTTGAG |
| cyclin D1 | TTTGTTGTGTGTGCAGGGAG | TTTCTTCTTGACTGGCACGC |
| CDK2 | TGAAGATGGACGGAGCTTGT | ACTGGAGGAGAGGGTGAGAT |
| MMP9 | GCGTCTTCCCCTTCACTTTC | ATAGGGTACATGAGCGCCTC |
| MAPK1 | GAACTTCTGCAACCCCACTG | CAGCCGCAGTTATAAGCAGG |
| N-cadherin | GACAATGCCCCTCAAGTGTT | CCATTAAGCCGAGTGATGGT |
| E-cadherin | CGGACGATGATGTGAACACC | TTGCTGTTGTGCTTAACCCC |
| HSP27 | AGTGGTCGCAGTGGTTAGG | TCCTTGGTCTTGACCGTCAG |
| c-Myc | AACACACAACGTCTTGGAGC | GCACAAGAGTTCCGTAGCTG |
| VIM | AGCTAACCAACGACAAAGCC | TTGCGTTCAAGGTCAAGACG |
| GAPDH | CGACCACTTTGTCAAGCTCA | AGGGGTCTACATGGCAACTG |

## Western blot analysis

The cultured cells were lysed with RIPA buffer (Beyotime, China). The protein concentration was quantified by Enhanced BCA Assay kit (Beyotime, China). Total protein samples were separated with 10% SDS-PAGE gel and transferred onto PVDF membrane (Millipore, USA). After blocked with 5% skim milk, the membrane was probed with primary antibodies against SNIP1 (1:1,000, Proteintech, USA) and GAPDH (1:20,000, Proteintech, USA) at 4 °C overnight and followed by incubation with secondary antibodies (1:4,000, Beyotime, China). Blots were visualized by BeyoECL Plus Kit (Beyotime, China) and scanned with a ChemiDoc XRS imaging system (Bio-Rad, USA).

## Dual luciferase reporter assay

To determine the binding affinity between SNIP1 and miR-29a-3p, the recombinant psiCHECK-2 vectors (Promega, USA) with the wild type (WT) or mutant of SNIP1 gene 3′-UTR were constructed. Then, the recombinant vectors (WT or mutant) and miR-29a-3p mimics (or NC) were co-transfected in HeLa cells. Dual-Luciferase Reporter Assay system (Promega, USA) was used to calculate the luciferase activity after transfection according to the manual instruction.

## Scratch assay

Transfected HeLa cells were cultured in 6-well plates until the confluence reached 100%. Then, a sterile pipet tip was used to scrape on the bottom of culture plates. Cell migration was observed at 12 h under an inverted microscope (Olympus, Japan) and images were captured for each sample. The scratch area was measured and analyzed by ImageJ software (NIH, MD, USA).

## Cell Counting Kit-8 assay

Hela cells (3,000 cells/well) were seeded into 96-well plates after transfection, and cultured for 24 h, 48 h, 72 h and 96 h. Fresh medium with 10% CCK-8 (Genview Scientific, AUS) was mixed carefully, and the absorbance values of 450 nm wavelength were detected at least three times by a spectrophotometric plate reader (Hitachi, Japan).

### Transswell migration assay

Transfected cells ($5 \times 10^4$) were placed into the top chamber of the transwell inserts (Corning, USA) and maintained with serum-free medium. Then the lower chamber was filled with complete medium. And the migrated cells were fixed and stained 24 h later. Finally, the cells stained in more than three visual fields were randomly selected and photographed with an inverted microscope (Olympus, Japan) and counted.

### Statistical analysis

Statistical analysis was carried out using the GraphPad Prism 8 (GraphPad Software, SanDiego, USA) program. All results were presented as mean ± standard error (SD) of three independent experiments. Comparison among multiple groups was assessed by Student's $t$ test. A statistically significant difference was defined as $P < 0.05$.

## RESULTS

### Knockdown of SNIP1 reduced migration and proliferation in cervical cancer cells

To address the biological role of SNIP1 in the progression of cervical cancer, three small interference RNAs (siRNAs) targeting SNIP1 were synthesized and transfected into HeLa cells. Among them, siSNIP1-330 showed the best silencing effect (Figs. 1A and 1B), which was subsequently chosen for further analysis. After transfected with siSNIP1-330, the wound area ratio increased (Figs. 1C and 1D) and the number of migrated cells decreased (Figs. 1E and 1F), which demonstrated the migration of HeLa cells was suppressed. Furthermore, the cell proliferation rate declined after transfection for 48 h (Fig. 1G). In addition, the mRNA expression levels of several migration-related genes (*MMP9*, *VIM*, *MAPK1*, *N-cadherin* and *E-cadherin*) and proliferation-related genes (*CDK2*) in HeLa cells (*Wang & Chen, 2019*) can also be downregulated or upregulated (Fig. 1H). Hence, SNIP1 knockdown could reduce migration and proliferation in cervical cancer HeLa cells.

### SNIP1 was directly targeted by miR-29a-3p

To further confirm the exact miRNA that can directly target SNIP1, three different bioinformation tools starBase, TargetScan and miRanda, were performed to calculate the possibility scores. Intersection of these three sets showed that there were 12 candidate miRNAs which target SNIP1(Fig. 2A), and miR-29a-3p was chosen for the highest score (Table 2). Moreover, the expression relationship between miR-29a-3p and SNIP1 is negative correlation ($P < 0.05$) in cervical squamous cell carcinoma and endocervical adenocarcinoma (CESC) samples from starBase (Fig. 2B, Table 2). Compared with the control group, the luciferase activity was inhibited significantly in HeLa cells when co-transfected with SNIP1 wild type 3′-UTR vector and miR-29a-3p mimics (Figs. 2C and 2D). Furthermore, both mRNA and protein expression levels of SNIP1 in HeLa cells were declined after transfection with miR-29a-3p mimics (Figs. 2E and 2F). Taken together, these results demonstrated that miR-29a-3p targeted SNIP1 via directly binding its 3′ UTR region and negatively regulated SNIP1 expression in cervical cancer.

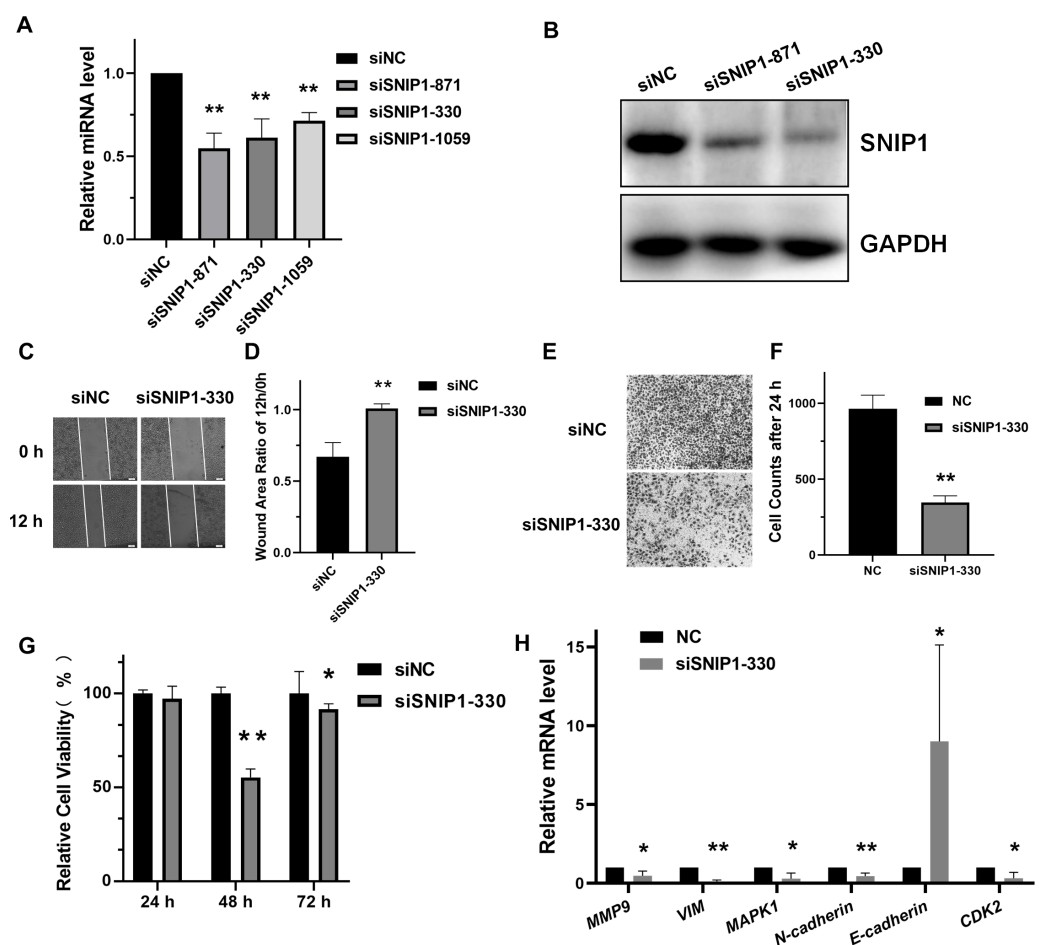

**Figure 1** **Knockdown of SNIP1 inhibited HeLa cells migration and proliferation.** (A–B) RT-qPCR and Western blot analysis of SNIP1 expression in different siRNAs transfected HeLa cells. (C–F) Scratch assay and transwell analysis were performed to determine the migration of HeLa cells transfected with siSNIP1-330 or siNC, respectively. (G) Cell viability was measured using the CCK-8 assay after knockdown SNIP1 in HeLa cells at 24 h, 48 h, and 72 h. (H) RT-qPCR analysis of migration-related genes and proliferation-related genes in HeLa cells infected with siSNIP1-330 or siNC, respectively. GAPDH was used as internal control. Each experiment were repeated three times, $*P < 0.05$, $**P < 0.01$.

## MiR-29a-3p inhibited migration and proliferation in cervical cancer cells

To evaluate the regulatory roles of miR-29a-3p in HeLa cells, the scratch assay and transwell assay for migration were performed. As results shown in Figs. 3A, 3B, 3D and 3E, transfection with miR-29a-3p mimics restrained migration in HeLa cells. Moreover, miR-29a-3p mimics also significantly decreased the relative cell viability in HeLa cells (Fig. 3C). Additionally, miR-29a-3p also regulated the mRNA expression levels of genes associated with migration or proliferation in HeLa cells (Fig. 3F). Therefore, these data indicated that miR-29a-3p inhibited migration and proliferation in cervical cancer cells.

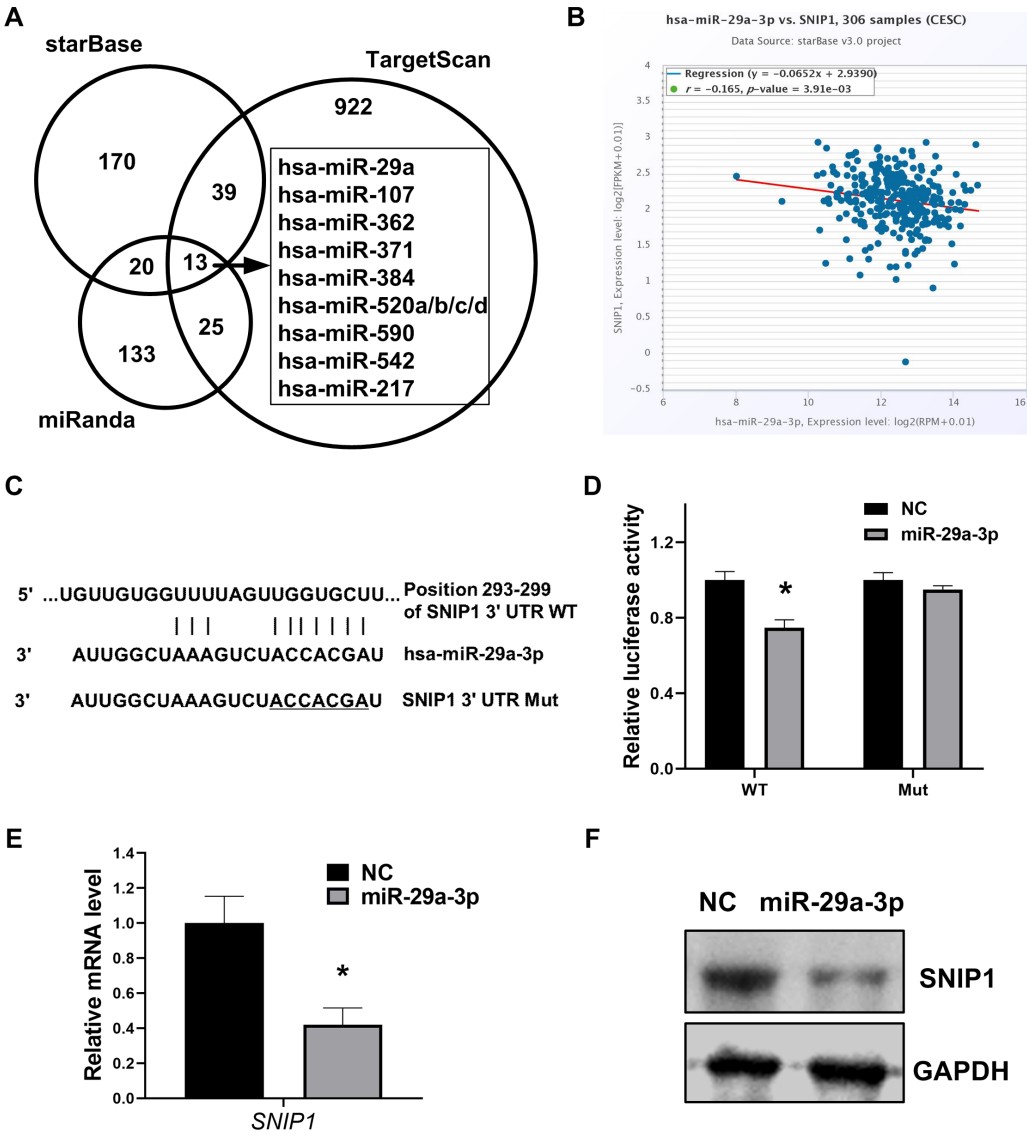

**Figure 2** **miR-29a-3p directly targets SNIP1 in HeLa cells.** (A) Potential miRNAs binding SNIP1 predicted in starBase, TargetScan and miRanda. (B) Correlation between miR-29a-3p with SNIP1 in CESC samples. (C) Graphical presentation of putative binding sites for miR-29a-3p in the wild type and mutant type 3′ UTR of SNIP1. (D) Dual-luciferase reporter assay was performed following transfection with miR-29a-3p mimics or NC mimics using psi-check2 luciferase vector including the SNIP1 WT 3′ UTR or the mutant 3′ UTR. (E) After transfection of miR-29a-3p mimics or NC mimics in HeLa cells, SNIP1 mRNA expression were detected by RT-qPCR. (F) SNIP1 protein expression was measured by Western blotting after transfection. GAPDH served as the internal control. *$P < 0.05$.

## MiR-29a-3p regulated the mRNA expression of SNIP1 downstream genes

To investigate whether miR-29a-3p would have effects on the downstream of SNIP1, the mRNA levels of these downstream genes (*HSP27*, *c-Myc* and *Cyclin D1*) (*Fujii et al., 2006*; *Bracken et al., 2008*; *Zhu et al., 2010*) were detected by RT-qPCR after transfection with

**Table 2  Potential miRNAs binding SNIP1 and co-expression analysis for the miRNA-SNIP1 in CESC.**

| miRNA | Position in the UTR[a] | Seed match[a] | Context++ score percentile[a] | r [b] | p-value[b] |
|---|---|---|---|---|---|
| hsa-miR-29a-3p | 293–299 | 7mer-m8 | 96 | −0.165 | 3.91E−03 |
| hsa-miR-542-3p | 836–842 | 7mer-m8 | 94 | −0.057 | 3.19E−01 |
| hsa-miR-384 | 314–321 | 8mer | 91 | 0.000 | 1.00E +00 |
| hsa-miR-520d-5p | 361–367 | 7mer-m8 | 90 | 0.023 | 6.91E−01 |
| hsa-miR-371a-5p | 504–510 | 7mer-m8 | 89 | −0.034 | 5.59E−01 |
| hsa-miR-520a-3p | 535–541 | 7mer-1A | 88 | 0.098 | 8.66E−02 |
| hsa-miR-520b | 535–541 | 7mer-1A | 87 | 0.012 | 8.29E−01 |
| hsa-miR-520c-3p | 535–541 | 7mer-1A | 87 | 0.017 | 7.64E−01 |
| hsa-miR-362-3p | 1,058–1,064 | 7mer-1A | 66 | −0.034 | 5.59E−01 |
| hsa-miR-590-3p | 1,191–1,197 | 7mer-m8 | 59 | −0.149 | 8.83E−03 |
| hsa-miR-107 | 2,140–2,146 | 7mer-1A | 46 | 0.013 | 8.15E−01 |
| hsa-miR-217 | 2,143–2,149 | 7mer-1A | 46 | −0.083 | 1.50E−01 |

**Notes.**

CESC, cervical squamous cell carcinoma and endocervical adenocarcinoma.

[a]Analysis from TargetScan7.2 (http://www.targetscan.org/vert_72/).

[b]Analysis from starBase v3.0 pan-cancer analysis (http://starbase.sysu.edu.cn/panMirCoExp.php).

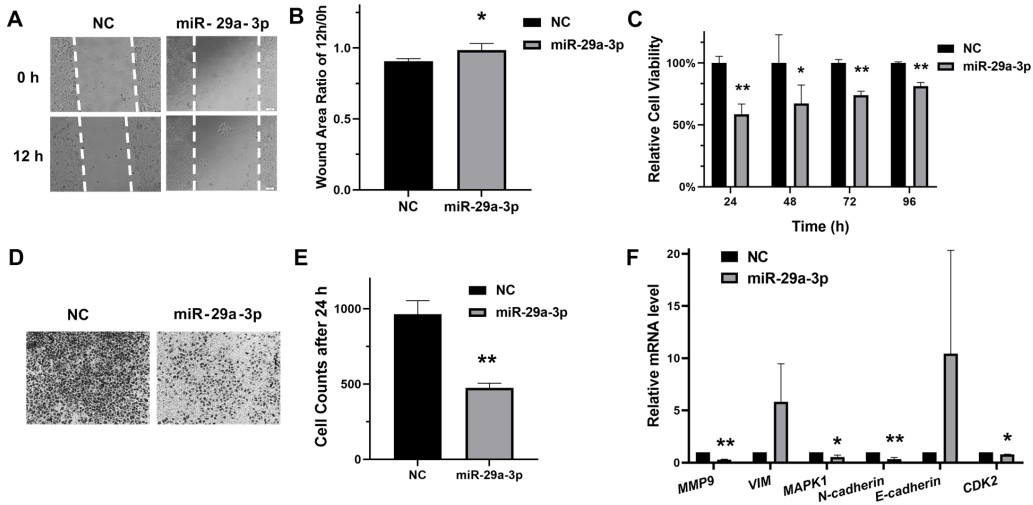

**Figure 3  MiR-29a-3p inhibited HeLa cells migration and proliferation.** (A–B) Cell scratch assay to evaluate HeLa cells migration after transfection of miR-29a-3p mimics or NC, respectively. The ratio of area 12 h/0 h was measured by Image J. (C) Cell viability was measured after transfection of miR-29a-3p mimics or NC. (D–E) The effects of miR-29a-3p on migration were determined by transwell assay in HeLa cells. (F) RT-qPCR analysis of migration-related genes and proliferation-related genes in HeLa cells transfected with miR-29a-3p mimics or NC. $*P < 0.05$, $**P < 0.01$.

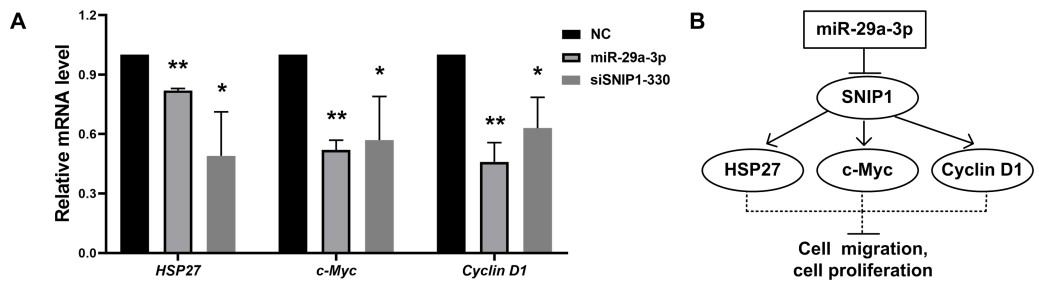

**Figure 4** **MiR-29a-3p regulated the downstream genes of SNIP1.** (A) The relative mRNA level of down-stream genes of SNIP1 (*HSP27*, *c-Myc* and *cyclin D1*) in HeLa cells transfected with miR-29a-3p mimics, siSNIP1-330 or NC were determined by RT-qPCR. (B) Schematic diagram of miR-29a-3p and SNIP1 regulate migration and proliferation in HeLa cells. $*P < 0.05$, $**P < 0.01$.

miR-29a-3p mimics or siSNIP1-330. The data suggested that miR-29a-3p and knockdown of SNIP1 both reduced the expression of those genes significantly (Fig. 4A).

## DISCUSSION

SNIP1 is a transcription regulator contains a nuclear localization sequence, and plays a key role in tumor development and progression (*Kim et al., 2000*; *Kim et al., 2001*; *Fujii et al., 2006*; *Bracken et al., 2008*). Over-expression of SNIP1 promoted cell invasion and migration in osteosarcoma cells (*Xie et al., 2019*). And the knockdown of SNIP1 restrained the anchorage-independent growth of lung cancer cells (*Jeon et al., 2013*). However, the function of SNIP1 in cervical cancer development is poorly understood. In our analysis, the migration and proliferation of cervical cancer cells was significantly suppressed after siRNA-mediated silencing of SNIP1 (Figs. 1C, 1D, 1E, 1F and 1G). These results suggested that SNIP1 was involved in the development of HeLa cells as an oncogene.

As a member of the miR-29 family, miR-29a is considered to play a crucial role in the regulation of multiple cancers (*Wang et al., 2018*). It was up-regulated and promoted epithelial-mesenchymal transition, migration and invasion in breast cancer cells (*Wu et al., 2019*). In contrast, other studies have shown that miR-29a was down-regulated and inhibited the progression of cancers such as colon cancer (*Shi et al., 2020*), non-small cell lung cancer (*Hu et al., 2016*) and adenocarcinoma (*Zhang et al., 2018*). Previous studies have shown that miR-29a was low expression in cervical cancer, as a tumor suppressor, miR-29a can target multiple genes, such as CDC42, HSP47, SIRT1 and DNMT1 (*Park et al., 2009*; *Yamamoto et al., 2013*; *Gong et al., 2019*; *Nan et al., 2019*). Nonetheless, the detailed biological function of miR-29a in cervical cancer has not yet been completely revealed. In this research, we showed for the first time that SNIP1 was targeted by miR-29a-3p in cervical HeLa cells. The results indicated that miR-29a-3p could down-regulate SNIP1 expression levels (Figs. 2E and 2F), and the direct binding site of SNIP1 mRNA 3′ UTR was confirmed by dual luciferase reporter assay (Fig. 2D). Furthermore, our data also supported previous reports (*Yamamoto et al., 2013*), indicating that miR-29a-3p can suppress the migration and proliferation of HeLa cells (Figs. 3A, 3B, 3C, 3D and 3E). Sharing the same seed region

in miR-29 family (miR-29a/b/c), we inferred that SNIP1 may also be targeted by other miR-29s.

It has been reported that SNIP1 could improve the transcriptional activity of c-Myc (*Fujii et al., 2006*), and regulate the stability of Cyclin D1 mRNA (*Bracken et al., 2008*). Otherwise, SNIP1 could down-regulate the transcription of HSP27 (*Zhu et al., 2010*). These downstream genes (c-Myc, Cyclin D1 and HSP27) regulated by SNIP1 are considered to be closely associated with cell proliferation and migration (*Evan et al., 1994*; *Zhu et al., 2010*; *Li et al., 2012*; *Pestell, 2013*). In this research, miR-29a-3p decreased the mRNA expression levels of *c-Myc*, *Cyclin D1* and *HSP27* in HeLa cells as an upstream regulator (Fig. 4A). Therefore, miR-29a-3p may mediate the regulation of cell proliferation and migration in cervical cancer cells via downstream genes of SNIP1 (Fig. 4B). Further study is required to illustrate the underlying mechanism of miR-29a-3p/SNIP1 pathway in cervical cancer oncogenesis.

In addition, this research has some defects in the following aspects. Although it was confirmed that miR-29a-3p can target SNIP1 to inhibit the migration and proliferation of HeLa cells, whether over-expression of SNIP1 could supplement the inhibitory effect of miR-29a-3p should be further observed. It was more convinced to detect the protein levels related to migration, proliferation as well as downstream of SNIP1, while the mRNA levels were reduced markedly by miR-29a-3p or SNIP1 siRNA in HeLa cells. In this study, we have used only one cell line to verify the function of miR-29a-3p/SNIP1, more cell lines and more in vitro and in vivo experiments are needed to be carried out. Finally, miR-590-3p is worth exploring as miR-29a-3p for the Pearson correlation analysis suggested that miR-590-3p was negatively correlated with SNIP1 in CESC tissues ($r = -0.149$, $P = 0.00883$) (Table 2).

## CONCLUSIONS

In conclusion, miR-29a-3p suppressed the migration and proliferation of cervical cancer cells by directly targeting SNIP1, and could also down-regulate the mRNA expression of SNIP1 downstream genes such as c-Myc, Cyclin D1 and HSP27. The newly identified miR-29a-3p/SNIP1 axis may provide new insights into the understanding of the progression of cervical cancer, and represent an effective treatment target for cervical cancer.

### Funding
This work was supported by the National Natural Science Foundation of China (No. 31601036). The funders had no role in study design, data collection and analysis, decision to publish, or preparation of the manuscript.

### Grant Disclosures
The following grant information was disclosed by the authors:
National Natural Science Foundation of China: 31601036.

## Competing Interests

The authors declare there are no competing interests.

## Author Contributions

- Ying Chen conceived and designed the experiments, performed the experiments, analyzed the data, prepared figures and/or tables, authored or reviewed drafts of the paper, and approved the final draft.
- Weiji Zhang and Lijun Yan performed the experiments, prepared figures and/or tables, and approved the final draft.
- Peng Zheng analyzed the data, authored or reviewed drafts of the paper, and approved the final draft.
- Jin Li conceived and designed the experiments, performed the experiments, analyzed the data, authored or reviewed drafts of the paper, and approved the final draft.

## Data Availability

The raw measurements are available in the Supplemental Files.

## Supplemental Information

Supplemental information for this article can be found online at http://dx.doi.org/10.7717/peerj.10148#supplemental-information.

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
