# Peer review of "miR-29a-3p directly targets Smad nuclear interacting protein 1 and inhibits the migration and proliferation of cervical cancer HeLa cells"

_PeerJ, doi:10.7717/peerj.10148_

## Round 0.1 · original submission · Minor Revisions

Please address concerns raised by both reviewers and revise your manuscript accordingly.

Reviewer 1 ·

Basic reporting

The research article by Chen et al. describes how miR-29a-3p directly targets Smad nuclear interacting protein 1 and reduces the migration and proliferation of cervical cancer HeLa cells. This is an important study as it opens up new avenue to target cervical cancer by inhibiting their proliferation and migration.

Experimental design

The experiments performed by the investigators are designed thoroughly and are interpreted clearly. There is sufficient details on the experimental procedures used and the author has clearly discussed the loopholes in the study.

Validity of the findings

However, I have certain queries and suggestions for the authors as given below:
1. I believe the title of the manuscript sounds vague. I would title this manuscript as “miR-29a-3p directly targets Smad nuclear interacting protein 1 is directly 1 and affects the migration and proliferation of cervical cancer HeLa cells”.
2. Figure 1E shows that the relative cell viability decreases significantly at 48 hrs post siSNIP1 treatment. However, at 72 hrs the viability seems higher than 48 hrs in the siSNIP1 treated groups. Please explain the possible cause of this effect?
3. Also, in the Figure 1F, E- Cadherin expression should be significant as the expression has increased by 10 folds post- siSNIP1-330 treatment and it lacks * mark on the figure.
4. Figure 2F blot is unclear and I would suggest it to be replaced with a clear one to see significant downreguation in SNIP1 expression at protein level.
5. Figure 3D again shows all markers significantly downregulated except E-cadherin which seems significantly upregulated statistically and lacks * mark.
6. Can you show the effect in vivo by injecting the control vs the miRNa transfected cells in a mice model to show the effect on HeLa cell migration and proliferation?

Additional comments

The manuscript definitely requires lot of corrections in terms of language and grammar. It lacks proper scientific writing, for eg: terms have been vaguely used such as in line number 135 “related genes (CDK2) in HeLa cells (Wang & Chen, 2019) can also be regulated (Fig. 1F)”. It can be rewritten as downregulated or upregulated.

Annotated reviews are not available for download in order to protect the identity of reviewers who chose to remain anonymous.

Reviewer 2 ·

Basic reporting

In this manuscript, Chen and colleagues report a novel regulatory interaction between Smad Nuclear Interacting Protein 1 (SNIP1) and miR-29a-3p with consequences on Hela cell proliferation and migration. Overall, I commend authors on doing a good job in putting together this concise and timely piece of work. However, I would recommend a few additional changes (also check Section 3) before acceptance, which I believe could help readers grasp and appreciate the scope of the study better.

A. The foundation of the entire study relies on HeLa cells but authors pitch the potential relevance of their study in context of cervical cancer. These claims should be toned down since authors don’t provide any data on levels of SNIP1 (and miR29a) from primary cervical cancer samples. While Hela cells are of cervical origin, at this point they have been growing in cultures for more than 5 decades, so it’s at best a cell culture model more than a cancer model.

B. Instead of the purported significance of the study to provide therapeutic solutions to cervical cancers, the highlight of the study could be the novel regulatory link between miR29a and SNIP and how this interaction plays a ‘master regulator’ role in controlling proliferation and migration. To me this is the most significant message from this work and authors could focus a bit more to highlight the same.

C. The manuscript is easy to read and generally written well. Introduction provides a good background of the miRNA and SNIP1 literature. But authors should provide more background on miR29a in the discussion.

Experimental design

As such, I find the primary research of this study lies within the Aims and Scope of Peer J. Regulation of SNIP1 is poorly understood and this work provides new mechanistic hints involving miRNA axis. The experiments are well planned and executed, adequate controls have been presented and raw data has been shared. Overall, the method sections provide sufficient details.

Validity of the findings

The findings are generally internally consistent with the underlying hypothesis and narrative, but I have a few concerns for authors to address:

A. In Fig 1E, viability of siSNIP1 drops significantly at 48h but recover to near-normal levels by 72h. Can authors comment or speculate on this peculiar trend. Is this because the effect of siRNA wears off by 72h. It would be good to support this finding by showing mRNA (and protein) levels at 48h v/s 72h for siSNIP1 scenario compared to control.

B. The figure legend for Fig 2 could be changed to ‘miR29a-3p directly targets SNIP1 to downregulate SNIP1 expression’. This change will reflect the data and its significance better for readers.

C. In Fig 3a, why do the NC condition also show poor migratory recovery in the wound-healing assay. Although, statistically significant, the effect size is very negligible between NC and miR29a case. Could authors comment on this?

Additional comments

The schematic in Fig 4B is not correct conceptually (at least to this reviewer). The relation between miR29a and SNIP1 (negative regulation, symbolised by ‘inverted T’) is shown correctly. But, SNIP1 is a positive regulator of HSP27, c-Myc and Cyclin D1 and they should be linked by ‘Arrow’ (indicating positive regulation) and not ‘inverted T’. Further downstream, arrows (or dashed lines) should link these 3 players to Cell Migration and proliferation. Dashed arrows/lines are best to use since authors don’t address any of these aspects in this study.

I hope authors will find these comments instructive and helpful in preparing their revised manuscript. If addressed suitably, I would recommend this manuscript for ‘acceptance’ at Peer J.

---

## Round 0.2 · accepted · Accept

Your clear and focused response to the comments of the reviewers is highly appreciated. I am satisfied with your answers and manuscript revision and have recommended your article for publication.